# Changes in Content of Polyphenols and Ascorbic Acid in Leaves of White Cabbage after Pest Infestation

**DOI:** 10.3390/molecules24142622

**Published:** 2019-07-18

**Authors:** Zuzana Kovalikova, Jan Kubes, Milan Skalicky, Nikola Kuchtickova, Lucie Maskova, Jiri Tuma, Pavla Vachova, Vaclav Hejnak

**Affiliations:** 1Department of Biology, Faculty of Science, University of Hradec Kralove, Rokitanskeho 62, 500 03 Hradec Kralove, Czech Republic; 2Department of Botany and Plant Physiology, Faculty of Agrobiology, Food and Natural Resources, Czech University of Life Sciences Prague, 165 00 Prague, Czech Republic

**Keywords:** *Brassica oleracea*, *Pieris brassicae*, *Phyllotreta* sp., phenolics, ascorbic acid

## Abstract

Crops, such as white cabbage (*Brassica oleracea* L. var. *capitata* (L.) f. *alba*), are often infested by herbivorous insects that consume the leaves directly or lay eggs with subsequent injury by caterpillars. The plants can produce various defensive metabolites or free radicals that repel the insects to avert further damage. To study the production and effects of these compounds, large white cabbage butterflies, *Pieris brassicae* and flea beetles, *Phyllotreta nemorum*, were captured in a cabbage field and applied to plants cultivated in the lab. After insect infestation, leaves were collected and UV/Vis spectrophotometry and HPLC used to determine the content of stress molecules (superoxide), primary metabolites (amino acids), and secondary metabolites (phenolic acids and flavonoids). The highest level of superoxide was measured in plants exposed to fifty flea beetles. These plants also manifested a higher content of phenylalanine, a substrate for the synthesis of phenolic compounds, and in activation of total phenolics and flavonoid production. The levels of specific phenolic acids and flavonoids had higher variability when the dominant increase was in the flavonoid, quercetin. The leaves after flea beetle attack also showed an increase in ascorbic acid which is an important nutrient of cabbage.

## 1. Introduction

Brassica crops, commonly known as crucifers, are grown worldwide for food and as animal feed and represent a significant economic value due to their nutritional, medicinal, bioindustrial, biocontrol, and crop rotation properties [1]. White cabbage (*Brassica oleracea* var. *capitata*) is a widely-cultivated crucifer vegetable with a high nutritive value due to its richness in active phytochemicals, such as vitamins C and E, carotenoids, minerals, dietary fiber, glucosinolates, phenolic acids, flavonoids, and anthocyanins [2]. White cabbage is extensively cultivated throughout the world, and the crop is often severely damaged by herbivorous insects such as the cabbage white butterfly, *Pieris brassicae*, and flea beetles which are the most common pests of the Brassicaceae family.

The large white butterfly (*Pieris brassicae* L.; Lepidoptera: Pieridae) has a life cycle lasting 45 days from egg to adult and there may be two to three broods per year with the first caterpillars hatching in spring (April to June), and the second during summer (July to August). Adults drink nectar of various plant species, while larvae exclusively feed on crucifers. Severe caterpillar infestations can destroy entire plants not only because of their feeding but also their feces. Cruciferous plants produce defensive compounds to repel insect attackers, but some larvae are able to safely accumulate the poisonous compounds in their bodies, which provides them protection from being eaten by some bird species. Because of this phenomenon, the cabbage butterfly has been used as a model species in the field or laboratory to study insect pest biology [1]. Flea beetles, primarily *Phyllotreta nemorum* and *P. undulata* (Coleoptera: Chrysomelidae), are the most common species on *B. oleracea* plants. They feed on cotyledons, young developing leaves, and stems of seedlings, leading to loss of photosynthetic capability and often to plant death. Feeding starts at the first two weeks after beetle emergence, and produces a shot-hole appearance and necrosis. Injury from larvae feeding on secondary roots hairs, however, caused a negligible effect on plant survival [3].

Plants have developed various defensive strategies against herbivorous insects. Apart from structural features like trichomes, thorns, or waxy leaf coatings, plants can reconfigure their metabolism to produce and accumulate specific chemical compounds that repel or even kill insects. These plant defense compounds act both as constitutive substances to repel herbivores through direct toxicity, or antifeeding properties by lowering the digestibility of plant tissues (e.g., by lignification), and as inducible substances produced in response to direct damage by herbivores. In addition, they may also play roles as antioxidants or as volatile attractants for predators [2,4]. In addition to the well-studied glucosinolate–myrosinase products, there are also a number of volatile compounds, such as lectins, phytoalexins, and phytoanticipins [5]. Phenolic compounds are common secondary metabolites in vascular plants. They exhibit great structural diversity, embodying a variety of functions in plant–herbivore interactions. They play important roles in pollination and oviposition, in host plant recognition by phytophagous insects, as feeding repellents, and in insect pest management [6]. Among the phenolics derived from phenylalanine are simple phenylpropanoids such as caffeic and ferulic acid, phenylpropanoid lactones (coumarins), and benzoic acid derivatives such as vanillin and salicylic acid [7]. The most common flavonoids in *Brassica* crops are quercetin, kaempferol, and isorhamnetin, commonly found as *O*-glycosides, mainly conjugated to glucose. They are also commonly acylated by different hydroxycinnamic acids [2].

Most of the published work has focused on the glucosinolate–myrosinase system, which is the best-studied chemical defense in crucifers [8,9]. Most research dealing with changes in phenolic metabolism has focused mainly on the absorption and sequestration of phenols and flavonoids in the bodies of *P. brassicae* caterpillars [10,11]. There have been no published reports about changes in phenolic metabolism after *Phyllotreta* attack. Therefore, the aim of this research was to study the changes in the accumulation of the main phenolic compounds in *Brassica oleracea* following attack by common insect pests. We focused on the herbivory of adult flea beetles (*Phyllotreta nemorum*) and the larvae of the large white cabbage butterfly (*Pieris brassicae*), specifically the direct feeding of 2nd instar larvae, the effects of oviposition, and the subsequent feeding of hatched caterpillars.

## 2. Results and Discussion

Plants have been generating complex defense mechanisms against various herbivorous insect feeding strategies over the entire long period of their evolution. Here, we studied the effects of adult flea beetle predation in the lab at two different levels, 50 or 100 insects per exposure (FB_50_, FB_100_), the feeding of 2nd instar larvae (WBC), or oviposition with subsequent feeding of hatched caterpillars of the large white cabbage butterfly (WBA). As soon as herbivore feeding starts on a plant, several defense signals are induced, leading to different defense responses. The injury by chewing of insects causes an immediate burst of reactive oxygen species (ROS), mainly hydrogen peroxide and superoxide radical, giving rise to both local and systemic responses [7]. In our study, the concentration of superoxide radical was elevated in nearly all plants exposed to insect predation. The presence of flea beetles, especially at lower numbers (FB_50_), as well as exposure to cabbage butterfly oviposition and larvae, WB_A_ and WB_C_, significantly increased the superoxide level in infested plants compared with controls (Figure 1a). Evidence from the literature, shows that the characteristics of the oxidative burst differ according to the type of plant and insect pest. For example, a strong accumulation of H_2_O_2_ was observed within 3 h of aphid infestation in wheat [12]. Additionally, histochemical staining of *Arabidopsis* leaves showed an accumulation of H_2_O_2_ 72 h after oviposition by *P. brassicae* [13]. In contrast, no ROS accumulation was observed in *Arabidopsis* plants up to 48 h after attack by the phloem feeding aphid, *Brevicoryne brassicae*. No evidence of ROS was found for up to 21 days after feeding of the sweet potato whitefly, *Bemisia tabaci*, however. A number of genes associated with oxidative stress, ROS scavengers, such as ascorbate peroxidase or catalase, were reported to be upregulated, which suggests that increased levels of ROS were not essential for triggering enhanced expression of oxidative defense genes, and secondary signaling pathways may be involved [14,15].

One characteristic plant response to insect attack is elevated protein content as a result of the induction of plant enzymes and nonenzymatic proteins involved in plant defense. Here, the content of total soluble proteins in injured leaves was not measurably affected by any of the used treatments (Figure 1b). In previous studies, contrary results were found in kale plants attacked by *P. brassicae* [16] and in corn plants infested with *Spodoptera frugiperda* [17], where a significant increase in total protein was measured. In general, phenolic compounds play a major role in host plant resistance to herbivores, including insects. Phenolics are synthesized in plants via the shikimic acid pathway. Phenylalanine ammonia-lyase (PAL) is the key enzyme catalyzing deamination of the aromatic amino acid, phenylalanine (Phe), to *t*-cinnamic acid, which participates in further reactions by conjugation with coenzyme A or hydroxylation to *p*-coumaric acid. Alternatively, *p*-coumaric acid is produced directly from tyrosine (Tyr), the hydroxyl derivative of Phe, via tyrosine ammonia-lyase, an analogue of PAL (Figure 2) [2,6,7]. The activity of PAL was strongly elevated in *Chrysanthemum* during the early period (0.5 to 6 h) after aphid infestation [18] and in kale after *P. brassicae* herbivory [16], and the enhanced PAL activity was correlated with elevated concentration of phenols. In our study, the activity of specific enzymes was not analyzed, but the levels of enzyme precursors were measured. The amount of Phe significantly rose only in plants attacked by flea beetles while the level of Tyr stayed unchanged. In contrast, herbivory by cabbage butterfly larvae resulted in a significant decrease in Phe and Tyr (Figure 1c,d).

Phenols possessing antioxidant activity are known to be produced by stressed plants. They may neutralize ROS directly or through enzymatic reactions. Their antioxidant activity depends on their chemical structure, the position and increased number of hydroxyl groups in the molecule leading to higher antioxidant activity. Even when phenolic compounds are oxidized by polyphenol oxidase or peroxidase to quinones, they can still be effective in defense reactions against herbivores [23,24]. On the other hand, glycosylation, the addition of a sugar moiety, results in lowering of this antioxidant activity [2]. In our study, plants exposed to flea beetles (FB_50_ and FB_100_) and hatching caterpillars (WB_A_) showed a significantly higher content of total soluble phenols in comparison with controls; but treatments did not differ between themselves. The WBC plants did not differ from controls (Figure 1e). In the literature, the described changes in the content of total soluble phenols are not consistent. Some studies reported higher phenols content [25,26] while others noted a decrease in their level [23]. Analysis of total phenols content is not only specific for molecules like phenolic acids or flavonoids, but other types of compounds can also be detected. However, it is still the most widely used method, especially in stress-related research, and values of particular metabolites are then usually determined in detail using more accurate methods, such as liquid chromatography or mass spectrometry.

Another commonly used analytical procedure in stressed plants is the evaluation of flavonoid content via spectrophotometric assay based on the formation of an aluminum chloride complex (sometimes expressed as AlCl_3_-reacted flavonols). Here, the content was significantly higher only in cabbage plants exposed to 50 flea beetles, FB_50_ (Figure 1f). Total flavonoid content includes amount of various molecules that are synthesized and further metabolized by subsequent reactions. Figure 2 shows various pathways leading to three individually analyzed aglycones—kaempferol, quercetin, and luteolin. In this study, the preceding flavonoid substrates for these compounds or their glycosides and other metabolites were not analyzed. Overall, the highest content was monitored for quercetin. Here, a significant increase compared to control was found for WB_C_ and FB_50_ plants. Surprisingly, hatching caterpillars (WBA) did not change the amount of quercetin. Total levels of kaempferol and luteolin were considerably lower in comparison with quercetin. Only flea beetle infestation resulted in significant enhancements in both compounds (Figure 3).

Beyond their well-known antioxidant properties, flavonoids also play an important role in insect–plant interactions. Studies of a variety of flavonoids have demonstrated their feeding deterrent and stimulant activity. The significant inhibition of feeding by flavone and dihydroquercetin, and the ability of apigenin or isorhamnetin to stimulate feeding suggest that the degree of predation of flea beetles on crucifers may depend on flavonoid structure. The most abundant cabbage flavonoids, quercetin, and kaempferol, showed only slightly elevated deterrent properties. However, a mixture of these flavonoids may influence feeding preferences [27]. There is evidence of selective uptake of flavonoids by *P. brassicae* larvae from their food sources (*B. napa*, *B. oleracea*) and their subsequent bioconversion to provide beneficial functions, such as protection from harmful UV radiation [10,11].

With regard to the phenolic acids participating in flavonoid biosynthesis, only *t*-cinnamic acid was strongly increased in plants subjected to herbivory by second instar larvae (WB_C_). Surprisingly, other insect exposures caused significantly lower values in comparison with controls (Figure 4a). The content of other phenolic acids, *p*-coumaric and caffeic, rose slightly after flea beetle attack, FB_50_, and both FB_50_ and FB_100_, respectively. The other two treatments led to a decrease in phenolic acid concentration, more so for caffeic acid (Figure 4b,c). These phenolic acids also serve as substrates for other compounds like ferulic or chlorogenic acid; but no correlation between herbivory and their content was observed. Moreover, their content dropped in all tested plants, with the exception of ferulic acid in plants exposed to one hundred flea beetles, FB_100_, where a significant increase was seen (Figure 4d,e). One possible explanation for this may be their involvement in lignin biosynthesis.

The phenolic acids, *t*-cinnamate and *p*-coumarate are also metabolized by β-oxidation to benzoate and 4-hydroxybenzoate, respectively. Benzoate can be further hydroxylated to salicylic acid which can serve as a signal molecule. The measured results, however, showed that insect-stressed plants did not have a significantly higher content of any of these compounds compared to controls (Figure 4f–h), but there were differences between individual exposures. Moreover, markedly lower amounts of salicylic and 4-hydroxybenzoic acid (Figure 4g,h) were detected in plants after flea beetle predation. Phenolics are known to play an antioxidant role in plant defense systems as a backup to the primary ascorbate-dependent detoxification system. In our study, the results showed a significant increase of ascorbic acid content after the flea beetles attack, while cabbage butterfly larvae herbivory showed only a weak response (Table 1). Few studies directly quantifying ascorbic acid content after damage of both chewing and sap feeding herbivores showed oxidation and loss of ascorbate in host plants [28,29]. In addition, enzyme activities and the transcript abundance of ascorbate peroxidase or oxidase, enzymes catalyzing generation of monodehydroascorbate and dehydroascorbate, were differentially modulated. Oviposition of *P. brassicae* downregulated a transcript encoding dehydroascorbate reductase [13], whereas larval feeding by a closely-related species, *P. rapae*, upregulated this transcript [30]. Surprisingly, the elevation of ascorbate content in plants by supplying them with its precursor enhanced the expansion rates of aphid colonies. One possible interpretation is that the excess ascorbate was utilized by aphids to enhance their metabolism. An alternative explanation is that an increased leaf ascorbate concentration decreased the lifetime of ROS signals in the plants and thus altered the balance of redox signaling pathways [31].

Oviposition and subsequent larval feeding represent a serious threat to crop plants. Thus, host plants have evolved direct defenses against egg laying (necrotic zones at the oviposition site) and indirect defenses such as the emission of volatiles to attract egg parasitoids or the synthesis of toxic or antifeeding compounds [32]. Geiselhardt et al. 2013 [8] showed that prior egg deposition of *P. brassicae* on *Arabidopsis* plants negatively influenced the feeding, growth, and survival of larvae but the concentrations of the major antiherbivory glucosinolates were not significantly increased by oviposition.

Treatment with the phenolic acids, *p*-coumaric, ferulic, salicylic, and protocatechuic, stimulated oviposition of *P. brassicae*, with the highest egg number after *p*-coumaric acid treatment. Here, the highest weight of feeding caterpillars was observed [33]. An antagonistic effect of other phenolic acids (syringic, coumaric, cinnamic, and vanillic acid) was found in castor plants. Their content rose after infestation with the castor semilooper, *Achaea janata*, or the tobacco cutworm, *Spodoptera litura*, and subsequent treatment of plants with these acids resulted in altered larval feeding preferences, where vanillic and cinnamic acids acted as repellents, and syringic and coumaric acids as attractants. The same preferences were also found for oviposition [34]. In our study, we determined only the changes in phenolic acid content. The oviposition of *P. brassicae* and subsequent larval hatching (WB_A_—exposure to adult cabbage butterflies laying eggs and larval hatching) led to a decrease of almost all monitored acids, significantly in the case of cinnamic and benzoic acid. A considerable increase in salicylic acid was monitored in extracts of cabbage plants subjected to oviposition and larval predation (Figure 4g). Its elevated level may point to a major role of salicylate in egg-induced plant responses. Oviposition as well as applications of extracts of *P. brassicae* eggs led to a strong accumulation of salicylic acid in *Arabidopsis* leaves [35]. The expression of several salicylic acid-responsive genes (PR1) was enhanced in leaf tissue beneath and close to *P. brassicae* eggs [13]. However, salicylic acid can be metabolized in *A. thaliana* to methyl salicylate. This volatile compound was not analyzed in this study and it could explain the difference between the production of benzoic acid metabolites and salicylates. Methyl salicylate acts as an attractant for the wasp, *Cotesia rubecula*, that parasitizes *Pieris rapa* caterpillars and its concentration was increased in the presence of herbivores [36]. The authors also stated that transcription of PAL was higher and they connected this fact with the induction of methyl salicylate biosynthesis. A similar mechanism could be involved in white cabbage, but this must be proven by further research.

## 3. Materials and Methods

### 3.1. Cultivation

Cabbage (*Brassica oleracea* var. *capitata* f. *alba*) seeds were sown in propagators using a mixture of multipurpose compost, garden soil, and perlite in a ratio of 2:2:1. Two weeks later, the seedlings were repotted into pots measuring 5 × 5 × 5 cm and left to grow for 25 days. Each 8-cell tray was placed in a 29 L transparent plastic box with a lid, and a thin layer of perlite was sprinkled on the box bottom. All boxes were equipped with two lateral ventilation windows measuring 10 × 15 cm, covered by metal nets, and a watering hose. During the whole experiment, the plants were cultivated in a Phytotron growth room (Weiss Technik) under the following conditions: 15-h photoperiod, temperature 22 °C (day)/17 °C (night), and humidity 60% (day)/70% (night). The plants were watered with an equal mixture of tap water and demineralized water.

Flea beetles (*Phyllotreta nemorum*) and large white cabbage butterflies (*Pieris brassicae*) (both adults and caterpillars) were captured on a cabbage field near Bolehost (50.2131900N, 16.0778428E, Czech Republic, 260 m a.s.l.), using aspirators for collecting the beetles, and entomological net bags for butterflies. On the same day, the insects were added into the cultivation boxes in the following way (treatments are bold):C—control (untreated plants)WB_A_—boxes with adult butterflies (five butterflies per box) to determine the influence of oviposition and feeding by hatched caterpillarsWB_C_—boxes with caterpillars (six caterpillars per box) in 2nd instarFB_50_—boxes with 50 flea beetles per boxFB_100_—boxes with 100 flea beetles per box.

The experiments were conducted as follows: The controls and the plants exposed to 2nd instar caterpillars and beetles were analyzed after four days; the plants exposed to adult butterflies were analyzed after eight days, after the females had laid eggs and new caterpillars had hatched. Plants were withdrawn from the boxes, leaves were gently cleansed with a brush and used for analysis.

### 3.2. Quantification of Stress-Related Compounds and Ascorbic Acid

Fresh leaves (0.1 g) were homogenized with 50 mM potassium phosphate buffer (pH 7.0) and centrifuged for 15 min at 14,000 rpm and 4 °C. The content of superoxide radical was determined by measuring nitrite formation from hydroxylamine (530 nm) [37]. A volume of 30 μL of supernatant and bovine serum albumin as the standard (595 nm) were used for determination of total soluble proteins [38]. All measurements were done with a Cintra spectrophotometer (Cintra 101, Dandenong, Australia).

The ascorbic acid assay was carried out as previously described by [39]. The leaves were kept at −18 °C between the harvest and the analysis. For each sample, 0.5 g of frozen leaves were homogenized with K_2_HPO_4_-PBS buffer (pH 2.5). The suspension was heated in a water bath to 75 °C for 45 min, cooled, and centrifuged (3000× *g*). The analysis was performed using an HPLC system (Agilent 1260 Series; Santa Clara, California, United States) with Kinetex C18 column (150 × 4.6 mm, 5 μm). The mobile phase was composed of a mixture of 97% 0.01 M K_2_HPO_4_-PBS (pH 2.5) and 3% methanol and was at isocratic elution with a flow rate of 1.0 mL min^−1^. The detection was at 210 nm and the concentration was calculated by comparison with a standard calibration curve.

### 3.3. Quantification of Phenols, Flavonoids, and Phenolic Acids

Extracts were prepared from fresh leaves using 80% methanol (*w*/*v* 0.2 g 2 mL^−1^). Total soluble phenols were estimated using the Folin–Ciocalteu method with gallic acid as a standard (750 nm). Total flavonoids (AlCl_3_-reacted flavonols) were quantified using AlCl_3_ as a reagent and quercetin was used as a standard (420 nm) [37]. All measurements were done with a Cintra spectrophotometer (Cintra 101, Dandenong, Australia). The content of phenolic acids and selected flavonoids was determined by UHPLC on a 2.1 × 50 mm, 1.8 μm Zorbax RRHD Eclipse plus C18 column (Agilent) with a 6470 Series Triple Quadrupole mass spectrometer (Agilent) (electrospray ionization—negative ion mode) as detector. Eluents: (A) 0.05% formic acid in water and (B) 0.05% formic acid in acetonitrile were used in the following gradient program: 0–1 min (5% B), 2.0–4.0 min (20% B), 8.0–9.5 min (70% B), 10.0–11.0 min (5% B). The MS source conditions were as follows: Gas temperature 350 °C, gas flow 9 L min^−1^, nebulizer 35 psi, sheath gas temperature 380 °C, sheath gas flow 12 L min^−1^, capillary 2500 V, and nozzle voltage 0 V. Selected MRM transitions were followed for each compound: 4-hydroxybenzoic acid (137.0 => 108.0, 92.0), benzoic acid (121.0 => 77.1), caffeic acid (179.0 => 135.0, 107.0), chlorogenic acid (353.1 => 191.0, 127.0), *t*-cinnamic acid (147.1 => 103.0, 77.0), ferulic acid (193.1 => 134.1, 178.0), kaempferol (185.1 => 1185.0, 239.0), luteolin (285.1 => 133.0, 151.0), *p*-coumaric acid (163.1 => 119.0, 104.9), quercetin (301.0 => 151.0, 179.0), salicylic acid (137.0 => 93.0, 65.0).

### 3.4. Quantification of Amino Acids

0.1 g of powdered dry leaves (80 °C for 24 h) was mixed with 1.5 mL of 70% ethanol containing 10 mM norvaline (as an internal standard). The suspension was blended for 15 min using a MultiReax shaker (Heidolph, Schwabach, Germany) and subsequently heated to 120 °C for 10 min. After cooling and centrifugation for 5 min at 4000 rpm, the supernatants were stored in Eppendorf tubes. The pellet was resuspended with 70% ethanol and shaken for another 15 min and again centrifuged. This process was repeated twice. The collected supernatants were centrifuged at 13,000 rpm for 3 min and heated to 60 °C under a N_2_ atmosphere (NDK 200-2, Hangzhou MIU Instruments Co., Ltd., Hangzhou, China) until they evaporated. Prior to analysis, the dry extract was dissolved in a mixture of mobile phases, 98% A and 2% B. The analysis was performed using an HPLC system (Agilent 1260 Series; Santa Clara, California, USA) with ZORBAX Eclipse Plus RRHT C18 column (50 × 4.6 mm, 1.8 μm) heated to 40 °C. Derivatization reagents (borate buffer 5061-3339, OPA Reagent 5061-3337) and amino acid standards 1 nmol μL^−1^ (5061–3330) were purchased from Agilent. Mobile phases, A (mixture of 10 mM Na_2_HPO_4_ and 10 mM Na_2_B_4_O_7_, pH 8.2) and B (acetonitrile/methanol/water 45/45/10) were run at a flowrate of 2.0 mL min^−1^ under the following gradient program: 0 min 2% B; 0.2 min 2% B; 7.67 min 57% B; 7.77 min 100% B; 8.3 min 100% B; 8.4 min 2% B; 9.0 min 2% B. The DAD (UV) detection was set at 338 and 390 nm for 0–6.1 min; 262 and 324 for 6.1–9 min and FLD detection was set at Ex 340 nm/Em 455 nm [40].

### 3.5. Data Processing

The values for the concentrations of the various compounds in control samples that were taken as 100% (Figure 1, Figure 3 and Figure 4) are given in Appendix A. The concentrations of the experimental samples were related to the 100% values. Each tested group was only compared with the compound content in the corresponding subculture. A mixed-model procedure, with a repeated statement for each parameter, was used to analyze the data set. Data from each measurement was tested separately. Fisher’s LSD test *(p* < 0.05) was used to determine significant differences. All statistical tests presented in this study were performed using a Statistica 13 (StatSoft Inc., Tulsa, OK, USA) software package. Principal component analysis (PCA); for establishing the effects of treatments on amounts of metabolite compounds was analyzed by the CANOCO 5 [41] software package.

## 4. Conclusions

In this paper, we showed that infestation of white cabbage by white cabbage butterflies or flea beetles caused changes in metabolism of stress compounds for both insect species. The results showed that these species had different effects on superoxide levels in predated leaves. The exposure to oviposition by butterflies and subsequent feeding by newly hatched caterpillars did not manifest in higher total phenolic content in comparison to predation by flea beetles. Despite the increased total flavonoid content in the case of the lower number of flea beetles, there was no clear prediction of which species could affect these secondary metabolites in principal component analysis (PCA) (Figure 5). The primary and secondary compounds like phenolic acids and flavonoids may play an important role in the defense against biotic stressors. Further detailed analysis of changes in individual phenolic metabolites are needed to explain their role in the defense response.

## Figures and Tables

**Figure 1 molecules-24-02622-f001:**
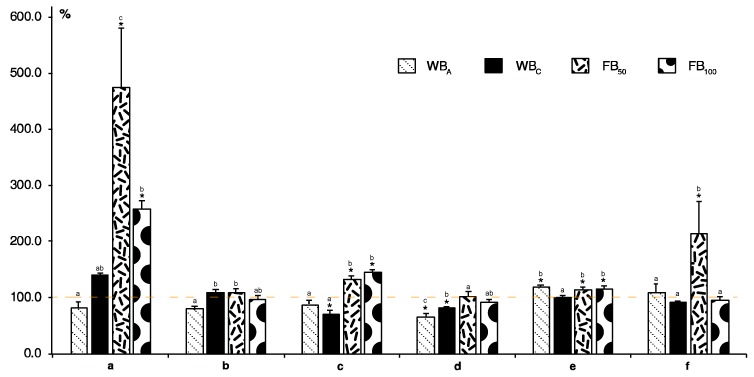
Effect of insect predation on production of stress compounds, and primary and secondary metabolites in white cabbage: (**a**) Superoxide, (**b**) total soluble proteins, (**c**) phenylalanine, (**d**) tyrosine, (**e**) total phenols content, and (**f**) total flavonoid content. WB_A_, adult cabbage butterflies (egg laying and hatching larvae), WB_C_ 2nd instar larvae, FB_50_ fifty flea beetles per box, and FB_100_, one hundred flea beetles per box. All bar values were recalculated relative to the compound content in untreated samples taken as 100% (dashed line). Data are means of three repeats ± SE. The asterisk (*) represents a significant difference between insect-damaged plants and controls, and different letters between the values of one compound. *p* < 0.05 by Fisher’s least significant difference (LSD) test.

**Figure 2 molecules-24-02622-f002:**
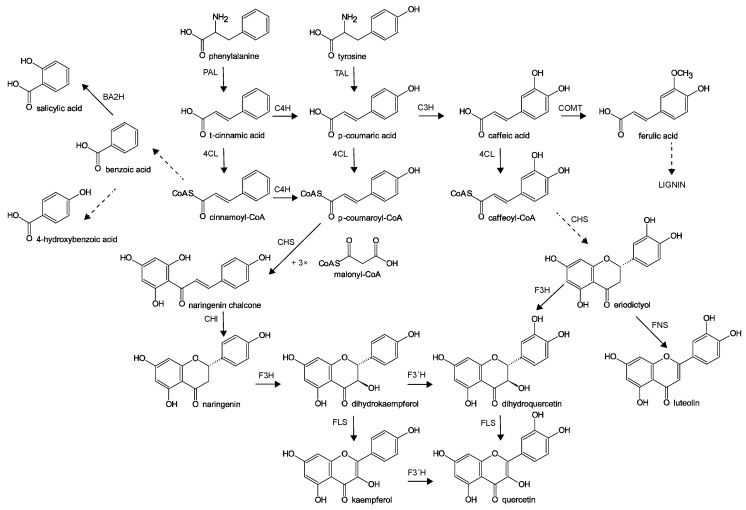
Pathways of phenolic acid and flavonoid metabolism [19,20,21,22]. PAL—phenylalanine ammonia-lyase; TAL—tyrosine ammonia-lyase; C4H—cinnamate 4-hydroxylase; 4CL—4-coumarate-CoA ligase; C3H—*p*-coumarate 3-hydroxylase; COMT—caffeic acid 3-*O*-methyltransferase; BA2H—benzoic acid 2-hydroxylase; CHS—chalcone synthase; CHI—chalcone isomerase; F3H—flavanone 3-hydroxylase; FLS—flavonol synthase; F3′H—flavonoid 3′-hydroxylase; FNS—flavone synthase.

**Figure 3 molecules-24-02622-f003:**
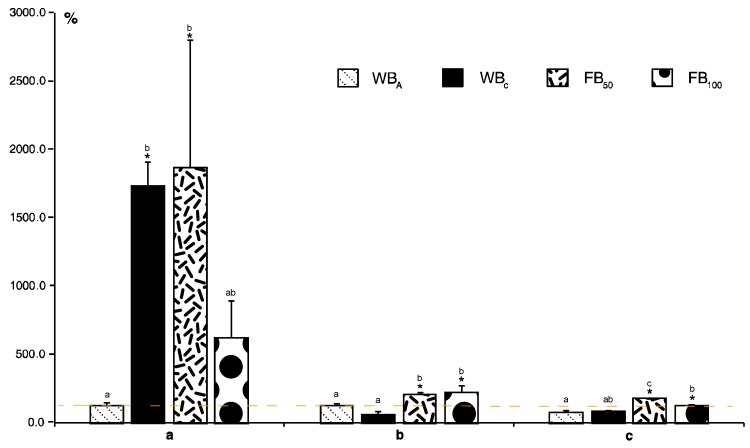
Effect of insect herbivory on flavonoid content in white cabbage. (**a**) Quercetin; (**b**) luteolin; (**c**) kaempferol. All bar values for experimental levels of compounds were recalculated relative to the content in untreated samples taken as 100% (dashed line). WB_A_, adult cabbage butterflies (egg laying and hatching larvae), WB_C_ 2nd instar larvae, FB_50_ fifty flea beetles per box, and FB_100_, one hundred flea beetles per box. Data are means of three repeats ± SE. The asterisk (*) represents a significant difference between treated samples and controls, and different letters between the values of one compound. *p* < 0.05 by Fisher’s least significant difference (LSD) test.

**Figure 4 molecules-24-02622-f004:**
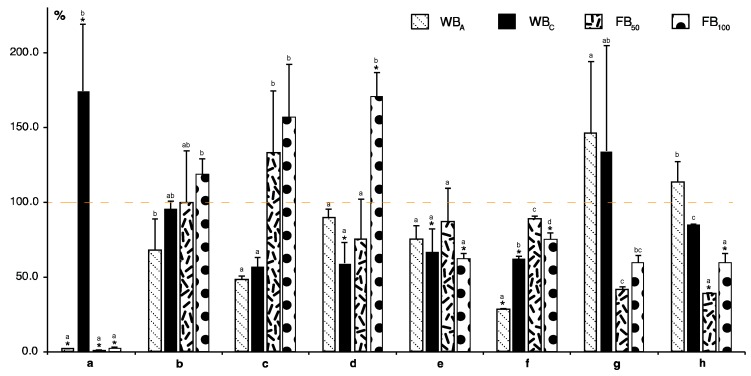
Effect of insect treatment on phenolic acids content in white cabbage. (**a**) *t*-cinnamic acid; (**b**) *p*-coumaric acid; (**c**) caffeic acid; (**d**) ferulic acid; (**e**) chlorogenic acid; (**f**) benzoic acid; (**g**) salicylic acid; (**h**) 4-hydroxy benzoic acid. All bar values were recalculated relative to the compound content in untreated samples taken as 100%. Data are means of three repeats ± SE. The asterisk (*) represents a significant difference between treated samples and controls, and different letters between the values of one compound. *p* < 0.05 by Fisher’s least significant difference (LSD) test.

**Figure 5 molecules-24-02622-f005:**
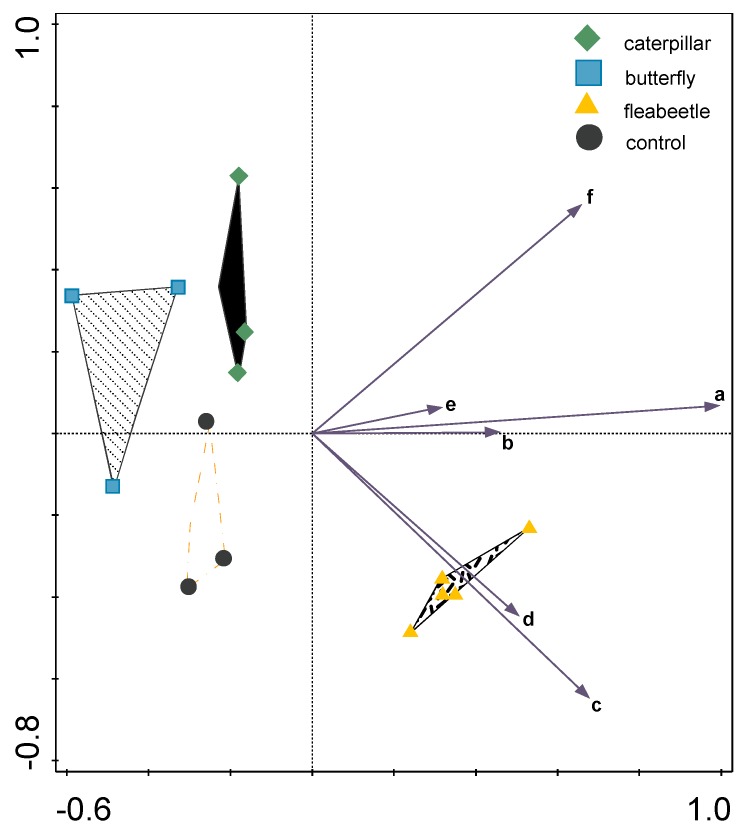
The principal component analysis (PCA) includes Figure 1 compounds and the Euclidean distances between particular samples. The total variation was 10.6. The explained variation on first two axes was 86.15%.

**Table 1 molecules-24-02622-t001:** The content of ascorbic acid (mg·g^−1^ FW) in extracts from white cabbage after predation by cabbage butterfly larvae or flea beetles.

Control	WB_A_	WB_C_	FB_50_	FB_100_
0.034 ± 0.003 ^a^	0.052 ± 0.004 ^a^	0.054 ± 0.013 ^a^	0.116 ± 0.006 ^b^	0.155 ± 0.033 ^b^

WB_A_, exposure to adult cabbage butterflies (egg laying and hatching larvae), WB_C_ 2nd instar larvae, FB_50_ fifty flea beetles per box, and FB_100_, one hundred flea beetles per box. Data are means of three repeats ± SE. Different letters between the values of each sample show significant difference at *p* < 0.05 by Fisher’s protected least significant test (LSD).

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
