# Peer review of "Changes in Content of Polyphenols and Ascorbic Acid in Leaves of White Cabbage after Pest Infestation"

_molecules, 2019, doi:10.3390/molecules24142622_

Round 1

Reviewer 1 Report

The aim of this manuscript was to study the changes in the accumulation of phenolic compounds depending on the invasion of insect pests, specifically oviposistion, hatching caterpillars and larval feeding of the large white cabbage butterflies and invasion of the flea beetle adults. The leaves infested by insects were evaluated by UV/Vis spectrophotometry and HPLC for content of stress molecules (superoxide), primary (amino acids) and secondary metabolites (phenolic acids and flavonoids). Although the reviewer appreciated authors′ efforts, the present results only displayed predictable observation and most of the performed analytical methods and targets were common. In conclusion, this manuscript is not recommended to be accepted for publication in the current format. Some major comments to be addressed as follows:

1.         The authors failed to highlight the scientific merit of this manuscript on the basis of this work. In another word, the authors should explain what the advancements are in scientific knowledge according to their data.

2.         There were some minor typographic, grammar, and format errors to be found in the text. Authors have to check and revise these errors.

3.         In the beginning of Results and discussion, authors have to illustrate the conditions of WBA, WBC, FB50, FB100, to make their discussion more clearly.

4.         Figure 2, authors have to present these structures in a more concise and clear manner.

5.         Figure 3, the unit of Y-axis should be presented. In addition, authors have to rationalize the comparatively high contents of quercetin.

6.         Figure 4 and related results, there were not significant variations observed among the detected phenolic acids. Therefore, how can authors make a conclusion that the primary and secondary compounds play an important role in defense reaction against these biotic stressors?

7.         In the References section, the writing manner of references did not follow the style of this journal strictly. Authors have to check and revise these errors.

Author Response

We are grateful to independent reviewers for valuable comments on the manuscript. We accepted all suggestions of reviewers, as indicated in revised version (*Molecules-529713_revAll_final.docx) of manuscript (we used "Track Changes" function in Microsoft Word) and below.

Reviewer #1: Review

1.       The authors failed to highlight the scientific merit of this manuscript on the basis of this work. In another word, the authors should explain what the advancements are in scientific knowledge according to their data. - The highlight were modified, amended in revised version.

2.       There were some minor typographic, grammar, and format errors to be found in the text. Authors have to check and revise these errors. – amended in revised version.

3.       In the beginning of Results and discussion, authors have to illustrate the conditions of WBA, WBC, FB50, FB100, to make their discussion more clearly - At the beginning of the Results and discussion caption is an explanation of the abbreviations of the used variants for better orientation in the upcoming text – all amended in revised version.

4.       Figure 2, authors have to present these structures in a more concise and clear manner.. – Completely new figure - amended in revised version.

5.       Figure 3, the unit of Y-axis should be presented. In addition, authors have to rationalize the comparatively high contents of quercetin. – Added and amended in revised version.

6.       Figure 4 and related results, there were not significant variations observed among the detected phenolic acids. Therefore, how can authors make a conclusion that the primary and secondary compounds play an important role in defense reaction against these biotic stressors? – New conclusions and amended in revised version.

7.       In the References section, the writing manner of references did not follow the style of this journal strictly. Authors have to check and revise these errors. – all amended in revised version.

Reviewer 2 Report

Although the paper shows some interesting hints, namely the evaluation of changes of some metabolites in cabbage leaves after pest infection, it also evidences some faults in the methodological approach and in the elaboration and discussion of the data.

As reported in: “3.3 Quantification of phenols, flavonoids and phenolic acids” the Authors evaluate:

i)          the total soluble phenols using the Folin-Ciocalteu method with gallic acid as a standard;  

ii)        total flavonoids using spectroscopic (AlCl3) method with quercetin as a standard

iii)     total phenolic acid (including kaempferol, luteolin and quercetin, that are no phenolic acids!) by UHPLC (LC/MS) method. Under what kind of chromatographic and MS conditions?

Why the Authors evaluated flavonoids in both methods? Similar results were obtained?

All these evaluated compounds are free aglycones present in the extracts or obtained after an hydrolyses procedure?

The Author use LC/MS method to evaluate some of metabolite present in the extracts. Why they only consider aglycones and do not evaluate the presence of the different glycosides (whole compounds)? With LC/MS approach it is possible to obtain much more precise and accurate information on the presence of true metabolites, as this approach normally used in this kind of investigation.

Based on the reported remarks, I think the paper cannot be accepted for publication.

Author Response

We are grateful to independent reviewers for valuable comments on the manuscript. We accepted all suggestions of reviewers, as indicated in revised version (*Molecules-529713_revAll_final.docx) of manuscript (we used "Track Changes" function in Microsoft Word) and below.

Although the paper shows some interesting hints, namely the evaluation of changes of some metabolites in cabbage leaves after pest infection, it also evidences some faults in the methodological approach and in the elaboration and discussion of the data. – amended in revised version

As reported in: “3.3 Quantification of phenols, flavonoids and phenolic acids” the Authors evaluate:

i)          the total soluble phenols using the Folin-Ciocalteu method with gallic acid as a standard; 

ii)        total flavonoids using spectroscopic (AlCl3) method with quercetin as a standard

iii)     total phenolic acid (including kaempferol, luteolin and quercetin, that are no phenolic acids!) by UHPLC (LC/MS) method. Under what kind of chromatographic and MS conditions?

We have included in the text an explanation of the use of mentioned analytical methods. In addition, in the Methods section, we supplemented the conditions of chromatographic and MS method. – amended in revised version

Why the Authors evaluated flavonoids in both methods? Similar results were obtained? AlCl3 method for the determination of total flavonoids is a commonly used analytical method for working with plant samples. This method quickly reveals changes in the accumulation of these metabolites, especially after the application of stressor. Within this method, a broad spectrum of flavonoids (predominantly flavonols) is determined. Subsequently we have determined by the UHPLC-MS method the concentration of the three most abundant flavonoids in cabbage – amended in revised version

All these evaluated compounds are free aglycones present in the extracts or obtained after an hydrolyses procedure?

All these evaluated compounds are free aglycones present in the extracts. We did not do a hydrolyses procedure.

The Author use LC/MS method to evaluate some of metabolite present in the extracts. Why they only consider aglycones and do not evaluate the presence of the different glycosides (whole compounds)? With LC/MS approach it is possible to obtain much more precise and accurate information on the presence of true metabolites, as this approach normally used in this kind of investigation.

In this study, the other preceding flavonoid substrates for these compounds or their glycosides and other metabolites were not analysed. Detailed analysis of changes in individual phenolic metabolites are the aim of further study.

Round 2

Reviewer 1 Report

The authors have tried hard to revise this manuscript. The revision is satisfactory, and thus it is recommended for publication.

Minor:

The word "content" in the title is suggested to be changed to "contents" 

Reviewer 2 Report

Corrections and clarifications have been included in the manuscript, which made it clearer especially in the Materials and Method section.

The revision is satisfactory so the paper can be published in the present form.